# Investigating Health-Related Features and Their Impact on the Prediction of Diabetes Using Machine Learning

Hafiz Farooq Ahmad [1], Hamid Mukhtar [2,*], Hesham Alaqail [1], Mohamed Seliaman [3] and Abdulaziz Alhumam [1]

1   Computer Science Department, College of Computer Sciences and Information Technology (CCSIT),
    King Faisal University, P.O. Box 400, Al-Ahsa 31982, Saudi Arabia; hfahmad@kfu.edu.sa (H.F.A.);
    hisham@mail.net.sa (H.A.); aahumam@kfu.edu.sa (A.A.)
2   Computer Science Department, College of Computers and Information Technology (CCIT), Taif University,
    P.O. Box 11099, Taif 21944, Saudi Arabia
3   Information Systems Department, College of Computer Sciences and Information Technology (CCSIT),
    King Faisal University, P.O. Box 400, Al-Ahsa 31982, Saudi Arabia; seliamanme@kfu.edu.sa
*   Correspondence: h.mukhtar@tu.edu.sa

**Abstract:** Diabetes Mellitus (DM) is one of the most common chronic diseases leading to severe health complications that may cause death. The disease influences individuals, community, and the government due to the continuous monitoring, lifelong commitment, and the cost of treatment. The World Health Organization (WHO) considers Saudi Arabia as one of the top 10 countries in diabetes prevalence across the world. Since most of its medical services are provided by the government, the cost of the treatment in terms of hospitals and clinical visits and lab tests represents a real burden due to the large scale of the disease. The ability to predict the diabetic status of a patient with only a handful of features can allow cost-effective, rapid, and widely-available screening of diabetes, thereby lessening the health and economic burden caused by diabetes alone. The goal of this paper is to investigate the prediction of diabetic patients and compare the role of HbA1c and FPG as input features. By using five different machine learning classifiers, and using feature elimination through feature permutation and hierarchical clustering, we established good performance for accuracy, precision, recall, and F1-score of the models on the dataset implying that our data or features are not bound to specific models. In addition, the consistent performance across all the evaluation metrics indicate that there was no trade-off or penalty among the evaluation metrics. Further analysis was performed on the data to identify the risk factors and their indirect impact on diabetes classification. Our analysis presented great agreement with the risk factors of diabetes and prediabetes stated by the American Diabetes Association (ADA) and other health institutions worldwide. We conclude that by performing analysis of the disease using selected features, important factors specific to the Saudi population can be identified, whose management can result in controlling the disease. We also provide some recommendations learned from this research.

**Keywords:** machine learning; prediction; feature importance; feature elimination; hierarchical clustering

## 1. Introduction

Diabetes mellitus (DM) is one of the most common chronic diseases worldwide. In 2019, the International Diabetes Federation (IDF) announced that the number of adults that are diagnosed with diabetes is approximately 463 million of the world's population [1]. In addition, IDF considers the Middle East as one of the highest regions in diabetes prevalence, and the World Health Organization (WHO) places Saudi Arabia as the highest among Middle Eastern countries [2] and fifth in the top 10 countries known for a high diabetes incidence rate in the world. It is expected that Saudi Arabia is heading to a higher position by 2035 [3]. The cost of medical treatment is also affected by the rapid growth of the number of individuals with diabetes, representing a large burden on government

health expenses. According to recent estimates, the cost of diabetes incurred by the Saudi government is at 17 billion Riyals and if those with glucose intolerance (pre-diabetes) progressed at the current observed rate, the total cost would be 43 billion Riyals [4] in the coming years. Besides, Saudi Arabia is known for its rapid growth in population and has encountered soaring economic development in the recent four decades, leading to lifestyle changes due to urbanization.

These changes have led to an increasing rate of chronic diseases. Many studies conducted to address the rapid growth of Diabetes Mellitus have either the objective to quantify the status of diabetics in the country [3,5], identifying the most frequently performed self-care behaviors [6], identifying factors related to diabetes control [7], or apply mathematical [8] or machine learning models for diabetes prediction [9]. All these efforts are related to the increasing demand to enhance healthcare quality and control the elevated growth rate of diabetes in the kingdom.

It is essential for federal or local governments to perform national or local screening and educate people through awareness programs. There is a need to invest in novel ways to prevent and help in the early detection of such an expensive disease [10]. Early prevention can limit the complications and their impact on the person's quality of life, resulting in a reduced cost with a positive impact on the community and the health system [11]. An upfront cost in the form of early investments by the governments can result in long-term benefits to the overall society.

We believe that the existing efforts may have their own benefits and usefulness in tackling the diabetes issues in Saudi Arabia however, there is a need to devise mechanisms for efficient, cost-effective, and easily-available solutions for diabetes identification in the general population. Given the constant rise in the diabetic population in the country, it is imperative not not only to identify diabetics from non-diabetic persons but at the same time, the factors associated with diabetes should also be delineated. By knowing and overcoming these factors, people can act to control them in time. Clinics and hospitals can also identify patients-at-risk by evaluating these factors.

Considering the above-mentioned context and the need for time, we are motivated to develop a solution for predicting diabetics from non-diabetic patients from the electronic health records obtained from local Saudi hospitals. Therefore, the goal of this research paper is to develop predictive classifiers and models to investigate real diabetic patients' data gathered from different Saudi hospitals and regions, utilizing different metrics. Although the obtained records have very few health-related attributes including lab test results such as cholesterol tests (HDL and LDL), and the diabetes-specific tests (FPG and HbA1c), our objective is to identify those factors that can be controlled by the patients-at-risk or non-diabetics as precautions for avoiding diabetes. Previous work in this direction has used a much larger number of variables in different contexts. Thus, the current work presents a novel perspective on diabetes prediction. The insights obtained from this work in the prediction of Diabetes Mellitus (DM) and its associated risk factors can be useful at different levels: To support and strengthen the existing findings of DM medical research, particularly, in the context of Saudi Arabia, to assist the community in understanding the causes and prevention of diabetes, and to help the government to allocate efforts in the right direction to minimize the effect of the growing number of diabetes patients.

With these objectives in mind, we developed a model that used five machine learning classifiers to evaluate two datasets; each one containing some basic attributes about the patients (age, gender, weight, height, presence of hypertension, and the level of physical activity), the results of cholesterol laboratory tests (HDL and LDL) and one of these two tests: HbA1c in one dataset and FPG in another dataset. With these limited sets of features, we evaluated each dataset and identified the importance of various features and their role in diabetes prediction. To improve the performance of our classifiers, we also applied feature elimination using feature permutation and hierarchical clustering techniques. Finally, using the rank-based correlation method, we also identified and analyzed correlation among various features and their impact on diabetes risk and prediction.

The remainder of this paper is structured as follows. In Section 2, we briefly explain DM followed by the related work done in diabetes classification. In Section 3, we explain our research methodology from data collection and preprocessing to the process of feature engineering and dataset creation. Section 4 explains the development of machine learning models for diabetes prediction and classification with a focus on improving the model performance through feature elimination. The results are then discussed in Section 5. Section 6 discusses the outcome and benefit of our research and Section 7 concludes this article.

## 2. Background and Related Work

### 2.1. Diabetes Mellitus (DM) and Risk Factors

DM is a set of endocrine disorders resulting in high levels of blood glucose in the human body due to a deficiency in insulin excretion or insulin action and sometimes both. It causes direct and indirect complications responsible for significant illness and death [5]. There are different types of diabetes, but the most common ones are Type 1 Diabetes (T1D) and Type 2 Diabetes (T2D). Type 2 (T2D) is the most common form of diabetes as around 90% of diabetic patients are T2D. The remaining 10% is classified as T1D or gestational diabetes, which may occur during pregnancies.

The blood test for the measurement of Hemoglobin A1c (HbA1c) level is clinically significant in prediabetes and diabetes diagnosis [12]. Similarly, the glucose in plasma of fasting subjects is accepted as a diagnostic criterion for diabetes [13]. Moreover, according to the American Diabetes Association (ADA) there is more harmony between blood tests such as FPG and HbA1c when compared to two types of blood tests in the separation of HbA1c [14]. Most of the existing work achieve good results for diabetes prediction only when they include these tests in their input to the machine learning model along with a myriad of other features [15–20]. However, as the number of features is reduced, such predictions cannot be made with greater accuracy, and in absence of these tests, it becomes impossible to identify the diabetic status of a patient with high certainty.

From a medical point of view, it is possible to avoid DM at an early stage or at least control its complications [21]. For example, individuals with a certain range of FPG and HbA1c, are considered as prediabetic patients [12]. Their early diagnosis can help in preventing their transition to becoming diabetic or in their recovery into the non-diabetic stage. However, identification of factors that can lead a person to transition to the status of diabetes in a population is a challenge, albeit some studies have identified factors such as hypertension or body size among some of the associated risk factors with diabetes [22]. Other studies have identified certain conditions that can only be determined through various blood or imaging tests [23–26]. The unavailability of such tests at most health facilities and the associated costs may prevent people from diagnosing with diabetes and, thus, a large part of the population remains undiagnosed until it is very late in the treatment process [27].

Despite these difficulties associated with the diagnosis of diabetes, prediction of diabetes using machine learning techniques has gained significant attention from the medical and informatics research community. Below, we identify some of the recent efforts in this direction.

### 2.2. Related Work

There are different Machine Learning (ML)-based methods for diabetes prediction as well as methods that use feature selection. We will review them next.

### 2.3. ML-Based Methods

Othmane et al. [28] applied and evaluated four ML algorithms (decision tree, K-nearest neighbors, artificial neural network, and deep neural network) to predict patients with or without type 2 diabetes mellitus. These techniques were trained and tested on two diabetes databases: One obtained from Frankfurt hospital (Germany), and the other one,

the openly available, well-known Pima Indian dataset (https://www.kaggle.com/uciml/Pima-indians-diabetes-database). These datasets contained the same features composed of risk factors and some clinical data such as the number of pregnancies, glucose levels, blood pressure, skin thickness, insulin, BMI (Body Mass Index), age, and diabetes pedigree function. The results compared using different similarity metrics give a classification accuracy of more than 90% and up to 100% in some cases. Similarly, many other approaches trained their models on similar features. For example, in [15–20]) the authors used the Pima Indian diabetes dataset by modifying the preprocessing steps, applying different algorithms and adjusting their hyperparameters to generate improved results. The limiting factor of these approaches is the inclusion of some features like skin thickness, insulin, and diabetes pedigree function, which are generally not available or recorded. Moreover, factors like skin thickness may result in the classification based on ethnic function, thus, preventing a wide-range applicability of the approach.

Lai et al. [29] built a predictive model to better identify Canadian patients at risk of having Diabetes Mellitus based on patient demographic data and the laboratory results. Their data included the patient features age, sex, fasting blood glucose, body mass index, high-density lipoprotein, triglycerides, blood pressure, and low-density lipoprotein. They built predictive models using Logistic Regression and Gradient Boosting Machine (GBM) techniques achieving good sensitivity results. But the authors did not mention their performance in accuracy or specificity, which usually has better sensitivity as a trade-off. Thus, their performance cannot be generalized. Like this, many research works have compared the performance of several machine learning using the selected metrics, while a different metric may give a poor performance on the same model. Many other approaches for diabetes classification concluded that a certain type of algorithms can give better results for prediction without considering the issue of the generality of their models [30,31].

A number of other studies have used the National Health and Nutrition Examination Survey (NHANES) (https://wwwn.cdc.gov/nchs/nhanes/) from the US Center for Disease Control (CDC) for the prediction of diabetes or other diseases. The NHANES data was initiated in 1999 and is growing every year in the number of records as well as the variables it considers in its surveys. These studies, while utilizing the main NHANES dataset, use some subset of variables for disease prediction or classification tasks. For example, Yu et al. [32] identified 14 important variables—age, weight, height, BMI, gender, race and ethnicity, family history, waist circumference, hypertension, physical activity, smoking, alcohol use, education, and household income—for training their machine learning models. Using two different classification schemes, they achieved 83.5% and 73.2% results for the area under the Receiver Operating Characteristic (ROC) curve. Semerdjian and Frank [33] added two more variables—cholesterol and leg length—in their analysis. By applying an ensemble model using the output of five classification algorithms they were able to predict the onset of diabetes with an AUC (Area Under Curve) of 83.4%. In both these studies, the number of variables (14 and 16) was significantly higher than would normally be available in most EHRs. Hospitals supporting the record of these variables may also not have the values for all these variables for maximum patients. This limits the generality or wide applicability of the approaches.

The study by Dinh et al. [34] used the NHANES dataset and various machine learning algorithms to predict variables that are a major cause for the development of diabetes and cardiovascular diseases. They also considered the prediction of prediabetes and undiagnosed diabetes. Logistic regression, support vector machines, random forest, and gradient boosting algorithms were used to classify the data and predict the outcome for the diseases. The authors used ensemble models by combining the performance of the weaker models to improve accuracy. In diabetes classification, they used 123 variables and achieved good prediction performance. A distinguishing aspect of their work was that the dataset was further categorized into laboratory dataset (containing laboratory results) versus non-laboratory (survey data only) dataset. Laboratory results were any feature variables within the dataset that were obtained via blood or urine tests. The purpose of the non-laboratory dataset was

to enable a performance analysis of machine learning models in cases where laboratory results were unavailable for patients, supporting the detection of at-risk patients based only on a survey questionnaire. According to their results, waist size, age, self-reported weight, leg length, and sodium intake were five major predictors for diabetes patients. The study found that machine learning models based on survey questionnaires can give automated identification mechanisms for patients at risk of diabetes. In non-laboratory data, the most important features included waist size, age, weight (self-reported and actual), leg-length, blood pressure, BMI, household income, etc. [34]. The exact number of variables used in non-laboratory data is not reported by the authors, and, thus, it cannot be concluded if their approach can be useful in general situations.

### 2.4. Feature-Based Methods

Feature selection has been used previously for improving the classification performance in different medical situations. Particularly, Matín-Gonzaĺez et al. have proved that by performing feature selection, the results of the classifier can be improved for the prediction of success or failure in Noninvasive Mechanical Ventilation (NIMV) in Intensive Care Units (ICU) [35]. Similarly, Akay [36] used F-score feature selection-based SVM model, and Chen et al. [37] used SVM with rough-set based feature selection for the improved diagnosis of breast cancer. Liaqat et al. [38] performed a premier study on developing deep learning-based classifiers for atrial fibrillation. They built six models based on feature-based approaches and DL approaches. However, their features are manually extracted while the DL methods are trained on raw data without any feature engineering, as they perform implicit feature selection. It is unclear how they performed the manual feature extraction, but manual feature extraction is not a preferred approach if this can be done automatically, as explained later in our case.

Amer et al. applied a feature engineering approach to gain clinical insight and, thus, improve the ICU mortality prediction in field conditions [39]. The authors used only linear hard margin SVM as it maximizes the separation between different classes. Feature selection was performed using statistical and dynamic feature extraction with an evaluation performed after each step. Any misclassifications after these two stages were investigated manually. A final phase of feature fine-tuning consists of seven steps and utilizes the vital signs as opposed to the selection of dimensions in the previous stages. Results were then obtained by evaluating the various combinations of feature selection performed in different stages. The interesting aspect of their approach is the combination of different features at different stages and improving the results step-by-step. A conclusion of the work was that different profiles of patients required a different set of features for efficiently predicting the mortality of patients.

Tomar and Agarwal used the hybrid feature selection technique [40] on three different datasets of diabetes, hepatitis, and breast cancer. Their model adopted Weighted Least Squares Twin SVM (WLSTSVM) as a classification approach, sequential forward selection as a search strategy, and correlation feature selection to evaluate the importance of each feature. In contrast, we applied permutation importance for feature selection, which is known to be a faster technique without the need for a selection strategy. Once the features were found, we could use any of the ML models to classify and predict the data.

Specific for the case of diabetes, Balakrishnan et al. used SVM ranking with backward search for feature selection in T2D databases [41], where they proposed a specific feature selection approach for finding an optimum feature subset that enhanced the classification accuracy of Naive Bayes classifier. Ephzibah [42] constructed a combined model using genetic algorithms and fuzzy logic for future subset selection. However, genetic algorithms have their own associated costs, and the proposed approach did not justify the cost compared to the achieved accuracy. On the same lines, Aslam et al. [43] used genetic programming with Pima Indian diabetes dataset by generating subsets of original features by adding features one by one in each subset using the sequential forward selection method. The approach is not only costly but their results using 10-fold cross validation with KNN

(K-Nearest Neighbor) and a specific configuration of genetic programming yielded an accuracy of about 80.5%, which is not up to par with other contemporary approaches.

Rodríguez-Rodríguez et al. [44] applied feature selection on T1D Mellitus (T1D) patients using variables like sleep, schedule, meal, exercise, insulin, and heart-rate. Using time-series data of these features and the Sequential Input Selection Algorithm (SISAL), they ranked features according to their importance with respect to their relevance in the blood glucose level prediction.

One approach for feature selection consists of clustering that has been mostly used for dimensionality reduction in text classification. But hierarchical agglomerative clustering that organizes features into progressively larger groups [45] have been used in structural classification. Ienco and Meo [46] used hierarchical clustering for improving the accuracy of classification on 40 datasets from the UCI Irvine[47]. Their experimental results show that the hierarchical clustering method of feature selection outperforms the ranking methods in terms of accuracy. On the diabetes data, they achieved an accuracy of 77.47% using the Naïve Bayes' classifier and 75.26% using the J48 based classifier. There are two limitations of their work. First, the accuracy is not as good as reported by other approaches on the same dataset. Second, the performance reported is only the accuracy of classification, but it is worse in other evaluations. Compared to their approach, we report a much higher performance in accuracy, precision, and recall scores.

Considering the above analysis, we conclude that most of the existing approaches (1) use features which are not generalizable, (2) use a large number of features that cannot be obtained in many real-world scenarios, (3) develop specific models that may not be generalizable, and (4) report only a specific metric for evaluation while ignoring other metrics that may have worse performance as an issue of trade-off between the various evaluation metrics. We approach the problem of diabetes prediction while considering these limitations.

In our approach, we use a minimum number of features, reducing them further by feature elimination. We apply five different classification models to avoid model-specific performance. We report that all the models performed equally well on the metrics of accuracy, precision, recall, and F1-score, implying that our data or features are not bound to specific models. Finally, our analysis also includes the identification of those factors that can have an indirect impact on the complications of diabetes.

## 3. Materials and Methods

We begin with data collection, preprocessing, feature engineering, and label assignment to explain how we obtain two different datasets from the same subset of features.

### 3.1. Data Collection

The anonymized Electronic Health Records have been acquired from five different Saudi hospitals across three regions: The Central region, the Western region, and the Eastern region. It contains data of around 3000 patients collected over two years from 2016 until 2018 through different departments such as outpatient, inpatient, and emergency. The obtained dataset consists of 16 features of numerical, binominal, polynomial, and date type. The initial features along with a brief description of each are listed in Table 1.

### 3.2. Data Preprocessing

In the data preprocessing phase, data is prepared to be suitable for cleansing and classification. The data is cleansed using normalization and transformation of some columns (features) for example, the birth date was used to generate the age of the patient. In addition, many patients were missing important feature values like Fasting Plasma Glucose (FPG) and Hemoglobin A1c (HbA1c). Since both features were used to initially classify a person as diabetic or non-diabetic, to establish the ground truth, all the instances that did not have these feature values were removed. As the number of missing features was very high, replacing the missing values for both features was not desirable. After filtering

the patients, our dataset decreased to 225 eligible patients for classification. However, 43 out of 225 patients were missing HDL and LDL values. HDL is considered as "Good Cholesterol"—higher HDL means better state—while LDL is considered as "Bad Cholesterol" therefore, lower LDL values are desirable. In the experiments, when HDL and LDL values were used, the records with missing values were dropped. FPG and HbA1c values were also transformed using the American Diabetes Association (ADA) as reference for the different value ranges [48].

**Table 1.** The set of features selected in our dataset for classification of diabetic and prediabetic patients.

| No. | Feature Name | Feature Type | Feature Description |
|-----|-------------|-------------|-------------------|
| 1 | Date of birth | Date | Values in date format |
| 2 | Gender | Binominal | F: Female, M: Male |
| 3 | Height | Numerical | Values in Centimeter (cm) |
| 4 | Weight | Numerical | Values in Kilograms (kg) |
| 5 | Hypertension (HTN) | Binominal | Values as Yes, No |
| 6 | Fasting Plasma Glucose (FPG) | Numerical | Lab test results measured in mmol/L |
| 7 | Hemoglobin A1c (HbA1c) | Numerical | Lab test results measured in percentage (%). |
| 8 | High-density lipoprotein (HDL) | Numerical | Lab test results in mmol/L |
| 9 | High-density lipoprotein (LDL) | Numerical | Lab test results in mmol/L |
| 10 | Physical Activity Level | Categorical | Values in L: Low, M: Medium, H: High |
| 11 | Diagnosis start date | Date | Values in date format |
| 12 | Primary diagnosis | Categorical | Values available in ICD10 code format. |
| 13 | Secondary diagnosis | Categorical | Values available in ICD10 code format. |
| 14 | Primary diagnosis full name | Categorical | Values indicate diagnosis full name |
| 15 | Secondary diagnosis full name | Categorical | Values indicate diagnosis full name |
| 16 | Region | Categorical | Values indicate the region of the patient whether in central, western or eastern region. |

### 3.3. Subject Exclusion

In this study, we excluded subjects whose age was less than 19 years to focus on the prediction of T2D by reducing the chances of T1D, which usually develops in children and adolescents. Previous work [32–34] also excluded similar data as well as data indicated as gestational diabetes, which is relevant to pregnant women however, since we lack information on pregnancy, we did not perform this step. By excluding these subjects, we were finally left with 162 instances.

### 3.4. Feature Engineering

Of the 16 features mentioned in Table 1, we had to apply techniques to modify some features to make them suitable for ML algorithms for improved classification. We proceeded as follows. The date of birth was replaced by the age feature. All the features containing diagnosis information (primary, secondary, and their full names) were removed as the diagnosis of patients included multiple diagnoses, most importantly T1D and T2D, and was removed to avoid leaking the classification information into the machine learning model. Finally, the region and diagnosis start date features were also removed.

After the initial feature selection, the dataset obtained consisted of 10 features: Age, height, weight, gender, Hypertension (HTN), Physical Activity Level (PAL), lab tests of Lipoprotein levels (HDL and LDL), Fasting Plasma Glucose (FPG) and Hemoglobin A1c (HbA1c). We would like to mention that age, height, weight, HDL, LDL, FPG, and HbA1c were all numerical features, while gender (M or F), HTN (Yes or No), and PAL (L, M, or H) were categorical features containing text or literals. As our implementation is done in the scikit-learn library (http://scikit-learn.org/), whose methods require numerical data for efficient processing, we converted the categories to numerical values. Instead of replacing text with numbers (e.g., L:0, M:1, H:2), we used one-hot encoding to prevent the implicit ordering caused by the numeric values.

At this stage, our data processing steps were finished. Before starting the analysis, it was imperative to identify each record as representing data for a diabetic or non-diabetic patient. In other words, each record was to be labeled with an appropriate class.

### 3.5. Label Assignment

The appropriate references to use for evaluating diabetes were the "Standards of Medical Care in Diabetes—2018" from the American Diabetes Association (ADA) [49] and considering the algorithm proposed by the American Association of Clinical Endocrinologists (AACE) [48]. Two medical experts were also consulted who guided in the diagnosis of diabetes including the factors related to predicting the development of diabetes among people. Their suggestions agreed with the ADA and AACE specifications. Based upon these criteria, any of the FPG or HbA1c laboratory tests could be used to classify patients into either a Diabetic (Y) or Non-Diabetic (N) class. Thus, we proceeded to create two different datasets based on the class labeling scheme. Using an algorithm, the data was automatically labeled in the datasets with either of these classes using the criteria.

#### 3.5.1. Dataset-1: HbA1c Labeling

In this case, a dataset was created by labeling each instance as diabetic if the value of HbA1c $\geq$ 6.5% (48 mmol/mol) otherwise it was classified as non-diabetic. This labeling resulted in n = 79 ($\approx$49%) instances as diabetic and n = 83 ($\approx$51%) as non-diabetic. We can see that the dataset is quite balanced.

#### 3.5.2. Dataset-2: FPG Labeling

In this case, a dataset was created by labeling each instance as diabetic if the value of FPG $\geq$ 126 mg/dL (7.0 mmol/L) otherwise it was classified as non-diabetic. This labeling resulted in n = 62 ($\approx$38%) instances as diabetic and n = 100 ($\approx$62%) as non-diabetic. Although the dataset with FPG labeling is not quite balanced as in the case of HbA1c labeling, it cannot be characterized as imbalanced either.

Thus, we get two labeled datasets with 8 common features (age, weight, height, gender, PAL, HTN, LDL, and HDL) and using one of the FPG and HbA1c features as input and the other as the label in each dataset. For convenience, we refer to these datasets as HbA1c-labeled and FPG-labeled datasets, where the HbA1c-labeled dataset contains FPG as an input feature and vice-versa.

## 4. Model Development

To analyze the effect of the choice of the HbA1c or FPG labeling attributes on the datasets with the remaining attributes common between the two datasets, we performed the task of diabetes prediction using five machine learning classifiers. Each classifier was evaluated against both datasets. The details of the classifiers and results of the predictions will be discussed in Section 5.

After getting the prediction results on the initial datasets, we planned on improving the results further by performing further analysis and evaluation through feature importance and feature elimination.

### 4.1. Feature Importance and Feature Elimination

Feature selection aims at filtering out features that may carry redundant information. It is a widely-recognized important task in machine learning with the aim of reducing the chances of overfitting of a model on a dataset [46]. There are several ways to select features for a model. One way is to use those features which are important in the predictive power to affect the classification accuracy. Based on the score assigned to each feature, its usefulness in predicting a target variable can be estimated. Many models provide an intrinsic mechanism to rank the features according to the value of their coefficients (e.g., in Support Vector Machines or Logistic Regression) or using the split-criteria (e.g., in

Decision Tree and Random Forest). The correlation between various features can also be used to discover more relevant and important features [50].

While classifiers like linear SVM and linear logistic regression are suitable for interpretation in the form of linear relationships among the variables, they fail to discover complex, non-linear dependencies in the data. Decision trees are suitable for finding interpretable non-linear prediction rules, but there have been some concerns about their instability and lack of smoothness [51]. Similarly, RF models are found to be biased by giving importance to categorical variables with a large number of categories [52,53]. More explicit and advanced mechanisms include the method of Recursive Feature Elimination (RFE) which provides the flexibility of choosing the number of features to select or the algorithm used in choosing the features [54]. The impact of an exploratory variable on a response variable is usually interpreted in isolation, this is usually inappropriately interpreted as an impact for business or medical insight purposes [55].

Permutation importance is one technique recently proposed for identifying measures of feature importance [53,55]. It is a reliable technique that directly measures variable importance by observing the effect on model accuracy by randomly shuffling each predictor variable. In addition, it does not rely on internal model parameters, such as linear regression coefficients, and can be used with other models such as those developed using RF.

Feature elimination aims to reduce the number of input variables when developing a predictive model. The objective is to remove the features that may be non-informative or redundant predictors in the model [56]. By reducing the input variables, not only is the computational cost of modeling reduced, but it may also result in improving the performance of the model. By eliminating weak predictors, we can also improve the generality of the model [55]. Although our datasets comprise a small number of features as well as a relatively small number of instances, we were more concerned with improving the performance of the models through feature selection and elimination. Thus, we applied permutation importance followed by hierarchical clustering to identify the features that could be eliminated.

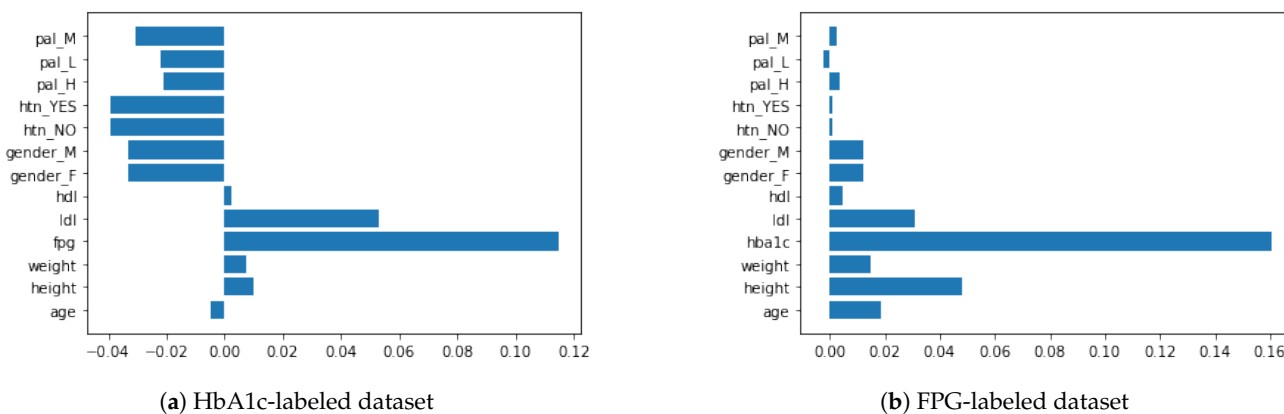

(**a**) HbA1c-labeled dataset    (**b**) FPG-labeled dataset

**Figure 1.** Permutation importance applied to the 9 features in HbA1c- and FPG-labeled datasets.

Figure 1 depicts the permutation importance applied to the two datasets. Although we can observe the importance of the FPG and HbA1c features in the HbA1c- and FPG-labeled datasets, respectively, the importance of the remaining variables is not consistent, given that they have all the eight features in common. This inconsistency is because the categorical features have been broken down using the one-hot encoding (as explained in the feature engineering subsection), resulting in collinearity among the features. For example, HTN (Yes/No) and Gender (Male/Female) are inversely correlated features. This is also evident in the case of Figure 1, where the collinear features have almost identical importance values. To avoid multicollinearity, as in our case, a strategy is to cluster features that are correlated and only keep one feature from each cluster. We applied Spearman's correlation by ranking the values of the variables and then running a standard Pearson's correlation

on those ranked variables as proposed by Parr et al. [53]. This resulted in a linkage matrix that is used to infer three main clusters, divided into further subclusters, as shown in the dendrograms for each dataset in Figure 2.

Compared to the permutation importance shown in Figure 1, the agglomerative hierarchical clustering in Figure 2 is consistent for both the datasets. Notably, the HTN=Yes feature is in the same cluster as the label (FPG or HbA1c) of the dataset, which implies they are close in their importance. Similarly, the height and Gender=M features are in the same cluster and are among the least predictive features. The final step is to flatten the cluster to its cluster components. By using the distance among the clusters as a criterion for cluster flattening, obtained from the linkage matrix computed earlier, we can identify the features that can be eliminated from the dataset. This process was applied to both datasets and Gender=M was the only feature that could be eliminated.

With one feature less than the total number of initial features, we performed the classification task once again. This is explained in the next section.

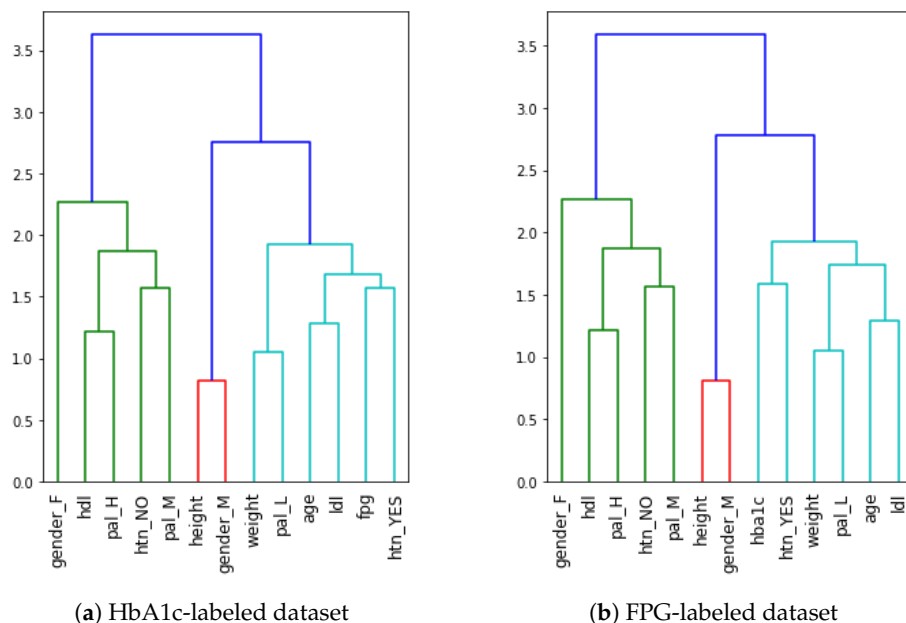

(**a**) HbA1c-labeled dataset　　　　　　　　　　　　　(**b**) FPG-labeled dataset

**Figure 2.** Dendrograms showing hierarchical clustering of the features in our datasets.

### 4.2. Selection of Machine Learning Classifiers

We chose five machine learning classifiers to evaluate the two datasets. These include three simple learners: Logistic Regression (LR), Support Vector Machines (SVM), and Decision Tree (DT) and two ensemble learners: Random Forest (RF) and Ensemble Majority Voting (EMV). RF uses a set of homogenous decision trees as its base classifiers while the EMV classifier was composed of the three simple learners LR, SVM, and DT, and used hard voting that considered the majority for predicting the class label for each instance in the test set. The rationale for choosing these is based on their previous performance reports in similar situations [9,31,32]. As our objective was to understand the factors contributing to the classification, we chose not to use any neural networks-based classifier in our analysis due to their "black-box" nature of interpretation of the model [34,57].

### 5. Results

For each dataset, two types of experiments were performed with all the classifiers. In the first experiment, all nine input features were used. In the second experiment, we performed feature selection and elimination before training and evaluating the classifiers, which resulted in eliminating one feature (Gender = M) from the dataset. With the eight final features, we performed the prediction task once again.

*Performance of Machine Learning Classifiers*

To measure the performance of each classifier, we used the widely-accepted performance statistics: Accuracy, precision, recall, and F1-score [58]. For model evaluation, we used 10-fold cross-validation in all experiments. The RF classifier used n = 100 estimators with max depth set to 40. Other parameters were left as default by the scikit-learn library. Both datasets were evaluated with the same model configurations. To allow reproducing the same splits across different experiments, we used the same seed for generating the random state for both datasets.

Table 2 and 3 describe the comparative performance of the five classifiers against each performance metric in the two experiments. The metrics represent the weighted average of the cross-validation. We learn the following from these tables:

- The performance of all classifiers was better in the FPG-labeled dataset as compared to the HbA1c-labeled dataset;
- SVM performed best on the HbA1c-labeled dataset while RF performed best on the FPG-labeled dataset;
- There was no change in the performance of SVM after feature elimination in both cases, while all the other classifiers saw an improvement or a decrease in the performance after feature elimination;
- The performance of DT and EVM classifiers improved, but that of RF decreased, after feature elimination in both cases.

The classification results are comparable to existing approaches for diabetes classification [9,31,32,34].

**Table 2.** Performance evaluation of the HbA1c-labeled dataset.

| | Evaluation with 9 Features | | | | Evaluation with 8 Features | | | |
|---|---|---|---|---|---|---|---|---|
| | Accuracy | Precision | Recall | F1-Score | Accuracy | Precision | Recall | F1-Score |
| Logistic Regression | 80.86 | 80.95 | 80.86 | 80.83 | 80.86 ↔ | 80.95 ↔ | 80.86 ↔ | 80.83 ↔ |
| SVM | **82.10** | **82.30** | **82.10** | **82.05** | **82.10 ↔** | **82.30 ↔** | **82.10 ↔** | **82.05 ↔** |
| Decision Tree | 74.07 | 74.07 | 74.07 | 74.06 | 75.31 ↑ | 75.34 ↑ | 75.31 ↑ | 75.28 ↑ |
| Random Forest | 81.48 | 81.91 | 81.48 | 81.38 | 80.86 ↓ | 81.61 ↓ | 80.86 ↓ | 80.70 ↓ |
| Ensemble | 77.78 | 78.14 | 77.78 | 77.66 | 78.40 ↑ | 78.86 ↑ | 78.40 ↑ | 78.26 ↑ |

**Table 3.** Performance evaluation of the FPG-labeled dataset.

| | Evaluation with 9 Features | | | | Evaluation with 8 Features | | | |
|---|---|---|---|---|---|---|---|---|
| | Accuracy | Precision | Recall | F1-Score | Accuracy | Precision | Recall | F1-Score |
| Logistic Regression | 83.33 | 83.31 | 83.33 | 83.04 | 82.72 ↓ | 82.62 ↓ | 82.72 ↓ | 82.45 ↓ |
| SVM | 84.57 | 84.74 | 84.57 | 84.22 | 84.57 ↔ | 84.74 ↔ | 84.57 ↔ | 84.22 ↔ |
| Decision Tree | 80.86 | 81.50 | 80.86 | 81.03 | 82.72 ↑ | 83.01 ↑ | 82.72 ↑ | 82.81 ↑ |
| Random Forest | **88.27** | **88.31** | **88.27** | **88.29** | 87.65 ↓ | 87.90 ↓ | 87.65 ↓ | 87.72 ↓ |
| Ensemble | 83.95 | 83.84 | 83.95 | 83.84 | 84.57 ↑ | 84.47 ↑ | 84.57 ↑ | 84.43 ↑ |

## 6. Discussion

Together with other features in the form of lab tests (LDL and HDL) as well as patient's information (age, gender, height, weight, hypertension, and physical activity level), we used HbA1c and FPG as features in two separate datasets for classifying an instance as diabetic or non-diabetic. In our experiments, we found that all five different classifiers predicted with better performance on the FPG-labeled dataset that included HbA1c as one of the input features. This implies that HbA1c can be used as a superior variable than FPG for diabetes prediction. This is consistent with the previous work as well. In a previous study on Vietnamese patients [59], researchers collected overnight fasting blood

samples from 3523 individuals (of which 2356 were women). Like our case, diabetes was diagnosed with an HbA1c value $\geq$ 6.5% or an FPG level $\geq$ 7 mmol/l. It was concluded that HbA1c testing had a higher sensitivity for identifying patients at risk for diabetes vs FPG, and therefore may have a greater impact on the diagnoses, cost, burden, and treatment of patients with diabetes [59].

When compared with the existing approaches, we can identify some distinguishing features of our approach. We used only a limited number of basic features (age, gender, height, weight, presence of hypertension, and physical activity level) and three laboratory tests (HDL, LDL, HbA1c, or FPG) to predict if a person has diabetes or not. In contrast, most of the existing approaches use many attributes. For example, Dinh et al. [34] initially used 123 features in diabetes prediction and even after removing the various laboratory tests, they were left with a much higher number of features (the exact number is not known). Finding these many features in real-world data is rarely possible. So, we proposed a mechanism whereby only with a few features could we infer the role played by them in the classification of a person into diabetic or non-diabetic.

The strategy for the identification of the contribution of each feature through the feature importance is also significant in the current analysis. Mostly, a correlation analysis is performed directly to identify such hidden patterns from data (e.g., as in [44]). However, as can be seen in Figure 1, visualizing the feature importance for different features does not reveal the same information as we have inferred from our results. Thus, we had to carry out a certain transformation in the form of clustering and distance evaluation to perform feature elimination. While we did not use correlation in the prediction task, the ranked correlations obtained during an intermediate step of our model development can be used to add to our findings.

### 6.1. Analysis of Diabetes Risk Factors

Figure 3a shows the correlation matrix of the initial dataset obtained after feature engineering, without applying any transformations, and Figure 3b shows the correlation matrix obtained after the ranked correlation based on the Spearman's ranking. While there have been small changes in some of the values, after the transformation, the correlation between the variables largely remains the same during the transformation process. Thus, the transformation has not affected the original relationship between the variables and the data remains integrated. The values of the correlation between some of the features of interest are shown in Table 4. We have organized the table into three sections: Lab results of HDL and LDL, hypertension, and personal attributes of age, weight, and height.

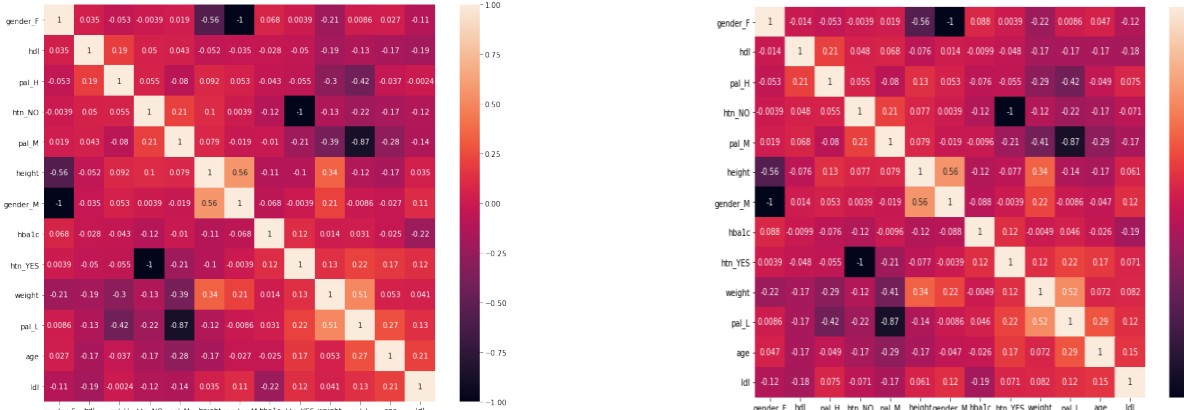

(**a**) FPG-labeled dataset before any transformations    (**b**) FPG-labeled dataset after going through the transformation process

**Figure 3.** Comparison of correlation before and after hierarchical clustering based on Spearman's ranking.

From the correlation analysis, we can establish the following information:

1. When we compare LDL to HbA1c and FPG, LDL is more correlated with HbA1c than FPG. On the other hand, HDL has almost no impact on HbA1c (close to 0);
2. The presence of hypertension is correlated with an increase in age as well as with a lower level of physical activity. Lower PAL is associated with hypertension while medium PAL is associated with the absence of hypertension;
3. Hypertension is also correlated with increasing levels of HbA1c and FPG, with an almost similar impact on both;
4. A higher level of physical activity has a good impact on HDL (the "good" cholesterol), while a low level of physical activity may cause higher levels of LDL (the "bad" cholesterol);
5. As the age of a person increases, so does LDL, meaning that younger people have comparatively small levels of dangerous cholesterol as compared to the older ones;
6. The level of physical activity decreases with the age. Thus, older people lack physical activities;
7. The level of physical activity of a person has also a strong correlation with the weight of a person, i.e., lower PAL indicates more weight while higher PAL is correlated with less weight of a person. Also, males have more weight when compared to females;
8. The height of a person is negatively correlated with both HbA1c and FPG. Accordingly, shorter people may be at higher risk of diabetes. This is also in accordance with existing findings [60,61]. In comparison, when we evaluate the weight of a person against HbA1c and FPG, there is almost no correlation between them (0.01);
9. There is no significant, direct relation between PAL and either HbA1c or FPG (<|0.05| in all cases of HbA1c and FPG with all PALs). Thus, we conclude that PAL has effects on weight, HTN, LDL, and HDL, which then have an impact on HbA1c and FPG levels, leading to diabetes.

These insights give us some hints into the diabetic disease, its development, and associated complications.

**Table 4.** Correlation between various features after ranking.

| LDL and HDL | | | Hypertension | | | Age, Weight, and Height | | |
|---|---|---|---|---|---|---|---|---|
| **Feature 1** | **Feature 2** | **Corr** | **Feature 1** | **Feature 2** | **Corr** | **Feature 1** | **Feature 2** | **Corr** |
| LDL | HbA1c | −0.19 | HTN=Yes | Age | 0.17 | Age | LDL | 0.15 |
| LDL | FPG | −0.11 | HTN=Yes | PAL=L | 0.22 | Age | PAL=L | 0.29 |
| LDL | PAL=L | 0.12 | HTN=No | PAL=M | 0.21 | Weight | PAL=L | 0.52 |
| HDL | PAL=H | 0.21 | HTN=Yes | HbA1c | 0.12 | Weight | PAL=H | −0.29 |
| HDL | FPG | −0.14 | HTN=Yes | FPG | 0.13 | Weight | Gender=M | 0.22 |
| HDL | HbA1c | −0.01 | | | | Height | HbA1c | −0.12 |
| | | | | | | Height | FPG | −0.12 |

*6.2. Recommendations*

With insights from the current work, we can present some recommendations. First, we can see that even with limited data, patients can be pre-screened for diabetes, and in case of their classification as diabetic, they can be advised to make appropriate changes to their lifestyle. We have found that physical activity plays an important role in diabetes development. Lower levels of physical activity were found to correlate with more weight, higher levels of LDL (the "bad" cholesterol), as well as hypertension. Thus, everyone needs to include higher levels of physical activity in their daily routines and avoid associated complications. Second, LDL has been found to have an association with HbA1c and FPG and the LDL levels also increase as a person progresses in age. In a similar fashion, HDL

levels are associated with FPG. Thus, it is important that people perform regular LDL/HDL tests and to control their levels in case it increases with time.

In our current work, we only had access to the hypertension feature of a patient as being Yes or No. However in practice, the patient's blood pressure is recorded as diastolic and systolic values. Similarly, temperature, vision, waist size, etc. are some other features that can be recorded with commonly available instruments in every clinic. Thus, just like age and weight play an indirect role in diabetes prediction, these and other features may also play a certain role in diabetes prediction and should be recorded for each patient to improve the diagnostic process. Finally, the government could enforce the pre-screening of diabetes based on age, weight, physical activity levels, and the presence of hypertension. These factors do not require any specialized tests or equipment and can be checked in any clinic, even in rural areas. By controlling these basic factors, a large segment of the population can be averted from developing early diabetes, a problem that has a large economic and social burden in many countries including Saudi Arabia. It is also important that accurate recording of physiological data should be enforced by hospitals and local clinics for any visiting patients for better opportunities to diagnose patients-at-risk. The data should be recorded in the patient's EHR so it can be used via a similar analysis on a larger scale to produce better analysis in the future.

### 6.3. Limitations of Work

We can also identify a few limitations within our work. First, as data availability is an important issue in health science research, although our data concerned 3000 patients, the final size of data was very small. The performance accuracy of a classification task mainly depends on the availability of large amounts of data [62] and with large data, we may have better insights. Unfortunately, our final dataset had only 199 records and after removing the missing values found for LDL and HDL features, we had only 162 records with complete feature values. With such small-scale data, there are limited options to test the available classifiers as well as the configuration of their various hyperparameters. That is why we did not invest time in further optimizing our classifiers for the given small dataset. With more data, better classifiers can be trained, evaluated, and optimized.

Second, the data were obtained in the context of Saudi Arabia. It would be interesting to test our approach to similar datasets from other countries/regions of the world. Third, in our current work, we used common machine learning classifiers. After establishing the feature importance of various features, we could even utilize black-box approaches like machine learning or deep learning and achieve state-of-the-art performance evaluation results [15–17]. This is one of our near-future goals.

### 7. Conclusions

The prevalence of diabetes is not only a burden for the governments in terms of the associated expenditures, but it is also a lifelong strain on diabetic patients. HbA1c and FPG are two important features for diabetes classification. With a dataset having both these important features for diabetes analysis, we constructed two separate datasets that classified an instance into diabetic or non-diabetic class. We found that HbA1c used as a future resulted in better performance (accuracy, precision, and recall) as compared to FPG. Moreover, we also identified several other features like hypertension, weight, and physical activity levels that had an indirect role in diabetic prediction. The LDL/HDL tests were also found to be correlated with diabetic conditions.

With data from other countries, our approach could be generalized, which may have important implications in the healthcare community. The prescreening of diabetes could be rapid, people could be more aware and educated about their lifestyles, and government expenditures could be reduced alongside the decrease in the significant burden on hospitals due to the prevalence of diabetes. With the ability to predict the onset of diabetes, necessary steps can be taken to avoid the diabetic stage of millions of people who are undiagnosed due to limited resources and lack of awareness. This can not only improve a person's quality of

life but also result in a positive impact on the healthcare system. Several recommendations have been proposed in this article in this regard.

**Author Contributions:** Conceptualization, H.F.A., H.A. and H.M.; methodology, H.M. and H.A.; validation, H.M. and M.S.; formal analysis, H.M., H.F.A., and M.S.; writing—original draft preparation, H.M., H.A., A.A.; writing—review and editing, H.M., H.A., H.F.A., M.S. and A.A. All authors have read and agreed to the submitted version of the manuscript.

**Funding:** This research was funded by the Deputyship for Research & Innovation, Ministry of Education in Saudi Arabia through the project number IFT20129.

**Institutional Review Board Statement:** Ethical review and approval were not required for this study, as it did not involve actual humans directly or indirectly in the study. The electronic health records obtained were provided anonymized by the source institutions and the study did not modify or applied any changes to the data. The research only anlayzed a set of data without any referral to any human.

**Informed Consent Statement:** Anonymized data without any reference to any patient was obtained and used in this research. The data cannot be traced back to the patients, so informed consent was not needed in this research.

**Data Availability Statement:** The authors choose not to make the data available yet. It might be available in the future.

**Acknowledgments:** The authors extend their appreciation to the Deputyship for Research & Innovation, Ministry of Education in Saudi Arabia for funding this research work through project number IFT20129. The authors also acknowledge the Deanship of Scientific Research at King Faisal University for the financial support Institutional Financing Track 2020.

**Conflicts of Interest:** The authors declare no conflict of interest. The funders had no role in the design of the study; in the collection, analyses, or interpretation of data; in the writing of the manuscript, or in the decision to publish the results.

## Abbreviations

The following abbreviations are used in this manuscript:

| | |
|---|---|
| EMR | Electronic Medical Record |
| SVM | Support Vector Machines |
| LR | Logistic Regression |
| DT | Decision Tree |
| RF | Random Forest |
| EVM | Ensemble Voting Model |
| DM | Diabetes Mellitus |
| T1D | Type 1 Diabetes |
| T2D | Type 2 Diabetes |
| HTN | Hypertenstion |
| PAL | Physical Activity Level |

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
