# Peer review of "Investigating Health-Related Features and Their Impact on the Prediction of Diabetes Using Machine Learning"

_applsci, doi:10.3390/app11031173_

Round 1
Reviewer 1 Report
See attached file.

Reviewer 2 Report
This manuscript is appropriately titled. The topic is relevant, timely, and would be of interest to the readership. The manuscript is not sufficiently well-written to merit publication. The ideas do not come together or flow logically to convey a clear message to readers. Its clarity and impact are lost in redundant unnecessarily wordy statements. The first paragraph of the introduction will almost certainly discourage readers’ interest. Readers might forego the effort to discern the nature, purpose and potential application of the case-finding model to strategically target follow-up clinical diagnostic and patient management resources in a more cost-effective and efficient risk-stratified manner.
Recommendations to authors for resubmission:
- enlist help from experienced writers in their academic department(s)
- focus your message, and keep the audience in mind. Those interested in this manuscript are unlikely to require the lengthy elementary description of diabetes mellitus.
- reduce sentences to concise statements of fact
- authors will need some assistance with English grammar and sentence structure
- the last sentence in section 6. Conclusions reads, “Several recommendations have been proposed in this article in this regard.” Those and the most important points readers can gain from your work should be itemized there. Don’t expect readers to go back through the manuscript to gather those recommendations.
- reconcile the references you’ve cited. Some seem superfluous and should be omitted. Cite only the most relevant literature.
Reviewer 3 Report
Authors apply machine learning techniques for a multi-class task on Diabete disease.
Particularly they analyze laboratory features, and study their predictive capability on the three classes Diabetic, pre-diabetic, and healthy.
The work is interesting, and experiments have been correctly conducted. The novelty of this paper is not the used methodologies (well known machine learning algorithms), rather they approach to the predictive task.
In the following some comments:
- Please proofread the article for typos and English grammar and style errors (e.g. predictation at page 2 row 62)
- page 4 row 155 authors state "In our view, all these efforts cannot be considered valid by the machine learning community". This is a very strong judgment that MUST be better justified. Otherwise I suggest to remove this sentence.
- Moreover, related work section should highlight the limits of the state-of-the-art approaches and how the proposed methodology overcomes them. I suggest to highlight what is the novelty of the proposed approach.
- Data: please define the data dimensionality. Moreover from table 3 it is clear that the dataset is strongly unbalanced since the healthy class is under-represented. I expect that because of this the classification models will fail in detecting that class. Thus, table 4 should include precision, recall and f1 measures for the three classes, in order to better understand how the algorithms behave with the unbalance class. If low values are returned, you should think about not considering that class, or use resampling methods to limit the problems.
- Experimental settings have not been detailed. Did you use fold validation? If not, you must re-run the experiments to get general results. How did you separated Training and Test sets? Moreover, you should detail the parameters of each algorithm that has been used (e.g. how many trees did you use for Random Forest?)
- Page 8, row 286: authors states that SVM could perform better if some configurations should have been applied, but they didn't. In my opinion you should make the classifier able to perform at its best, otherwise it is better if you do not use it. It os not correct to use SVM if you know that they perform bad because you didn't configure them properly.
- Feature importance is described in the last part of the article. The discussion is very interesting, however, you must describe how these features have been obtained.
- Table 4, case I. Since the labeling has been done based on the two features that are included in this dataset, I would expect to have 100% accuracy. How do you justify values lower than 100%?
Round 2
Reviewer 1 Report
As already indicated in the first round of review, the contribution of the work is not very significant and after the review it remains irrelevant because most of the suggestions for improvement have not been addressed.
The section of related works does not cover the aspects pointed out in the previous revision and has even suffered a reduction with respect to the first version.
Sometimes, instead of making the changes indicated in the revision, the authors have simply deleted the corresponding paragraph. This is the case of lines 148-155 of the first version, which have been deleted.
In other cases, instead of addressing the recommendations, they modify the objectives, simplifying the problem and reducing the relevance of the contribution. One of the most surprising answers of the authors regarding this issue is the following: “The data imbalance problem was solved by removing the prediabetes class”. The prediction of that class was among the main objectives of the work.
The authors themselves recognize that the objectives and context of the work have changed completely. This requires a new review of the whole work and not only of the parts indicated in the first round, so in that case the work should be considered as a new submission and not as a revision.
The last comment refers to the authors' reply document since they do not provide any details about the changes made or indicate in which lines the modifications are placed. In addition, the modified text is not highlighted in any way in the manuscript so it is very difficult to locate the improvements incorporated into the new version.
Reviewer 2 Report
The authors have transformed the manuscript.
The expanded methods and corresponding results credibly support the recommendations and conclusions.
The revision has rendered an article that is considerably more compelling, will be of greater potential impact, and the revised version is certainly more readable.
Author Response
We thank the reviewer for accepting this version.
The English mistakes will be checked once again thoroughly.
Round 3
Reviewer 1 Report
In the last version of the manuscript, the study of the state of the art has been considerably extended, including the discussion of the topic of feature selection, which has great relevance in the work developed. Additionally, the authors have more convincingly justified the change in their objectives. However, some minor changes are still needed:
- The sentence “The method of feature selection…” in line 183 is not appropriate since there are many feature selection methods, it must be rewritten.
- The paragraph between lines 343 and 352 does not fit the reality. On the one hand, this paragraph only mentions classification methods as techniques to quantify the importance of characteristics when the real methods to quantify such importance are feature selection methods. On the other hand, these methods have been widely applied in many fields, including medicine. Therefore, it is not true that the most frequently applied methods for quantifying feature importance are linear models and decision trees.
- The English of all the text that has been modified or added to the first version of the manuscript should be checked because there are numerous typos and incorrect phrases.
Author Response
We thank the reviewer for noticing and agreeing on the changes done in the second revision.
The sentence on line 183 has been updated as:
Feature selection has been used previously for improving the classification performance in different medical situations.
Lines 331-349 have been re-written/re-arranged to improve the flow and make it coherent.
The English grammar/spelling is improved. We made over 20 changes here and there in the text. Some examples:
unavailability of such test[s]
In spite of -> Despite
but it is clear that manual feature extraction -> but manual feature extraction
applied feature engineering approach in order to gain -> applied feature engineering approach to gain
seuqential -> sequential
a number of ways -> several ways
Other parameters were let as default -> Other parameters were le[f]t as default
use a large number of attributes -> use many attributes
Some sentences were reordered for better comprehension.
We hope the revision meets the standard of technical writing.
Once again, we thank the reviewer for his/her help throughout the manuscript submissions and appreciate the time taken out in a prompt manner.